# *Schizosaccharomyces pombe* in the Brewing Process: Mixed-Culture Fermentation for More Complete Attenuation of High-Gravity Wort

Barnaby Pownall, Struan J. Reid, Annie E. Hill and David Jenkins *

International Centre for Brewing and Distilling, School of Engineering and Physical Sciences, Heriot-Watt University, Riccarton, Edinburgh EH14 4AS, UK
* Correspondence: d.jenkins@hw.ac.uk

**Abstract:** High-gravity brewing is a method that maximises brewhouse capacity and reduces energy consumption per unit of beer produced. The fermentation of wort with high sugar content is known to impact the fermentation characteristics and production of aroma-active volatiles, and as such, cultures that are adapted to this method are industrially valuable. Mixed-culture fermentation offers brewers the opportunity to combine desirable features from multiple strains of yeast and to take advantage of the interactions between those strains. In this study, a highly attenuative strain of *Schizosaccharomyces pombe* is paired with a fast-fermenting brewing strain of *Saccharomyces cerevisiae* in the fermentation of wort at both standard and high gravity at centilitre scale. Mixed cultures were found to produce several esters and higher alcohols in higher concentration than in either of the parent monocultures at both standard and high gravity. The mixed culture also represented a compromise between fermentation length (modelled by the logistic equation), which was extended by the inclusion of *S. pombe*, and ethanol yield, which was increased. The application of mixed-culture strategies to high-gravity brewing practices may allow brewers greater flexibility in achieving desired flavour profiles whilst increasing brewhouse efficiency.

**Keywords:** beer; mixed fermentation; *Schizosaccharomyces*; high-gravity brewing; logistic modelling

## 1. Introduction

High-gravity (HG) brewing is an industrial technique employed by brewers to maximise their brewing capacity. Stronger worts—typically of around 16 to 20 °P (kg of extract per hL of wort)—are fermented, and the resulting beer is brought down to the target alcohol by volume (ABV) with deaerated, dechlorinated, microbially inactive water [1]. The higher end of this scale is often referred to as very-high-gravity (VHG) brewing [2]. HG methods are reported to increase the brewing capacity by between 20 and 30% [3]; thus, they represent potentially important efficiency savings for the brewer. High-gravity brewing is more sustainable and economical than standard-gravity brewing, due to the lower labour and energy costs per unit of packaged beer [4]. The fermentation of high-gravity worts places additional pressures on the yeast, which must perform under higher initial osmotic pressure, higher ethanol concentrations, and altered nutrient availability if adjuncts such as unmalted grain or refined sugars are added. The initial sugar concentration of the wort has been found to result in higher concentrations of acetate esters and higher alcohols [5], whilst the wort composition and availability of Free Amino Nitrogen (FAN) also impacts the production of these same compounds [6]. High-gravity production methods have also been linked with reduced post-fermentation viability and vitality of yeast [7], as well as comparatively poor head retention in the final product [8].

Accordingly, yeast strains that are capable of efficiently fermenting high-gravity worts whilst minimising the undesirable consequences of the technique are of significant commercial interest. Studies aiming to identify and develop yeast strains suitable for high-gravity

brewing have explored the improvement of existing strains by mutagenesis [9], adaptive evolution [10], and hybridisation [11]. Complementary to strain development strategies is the search for appropriate *Saccharomyces cerevisiae* strains from sources outside the brewing industry, such as the bioethanol industry [12] and cachaça production [13]. The fermentation of wort with high sugar content by non-conventional yeast strains—species that are not commonly employed in the brewing industry—has been investigated for both the bioethanol industry [14] and for winemaking [15] but remains an underexploited strategy for high-gravity brewing applications.

In extension to the use of non-conventional strains, mixed yeast cultures may find applications in HGB. A mixed culture consisting of a maltotriose-consuming strain of *S. cerevisiae* and a highly ethanol-tolerant strain of the same species was the subject of a 2013 patent [16], where the mixed culture produced a higher ethanol yield than either of the monoculture fermentations. However, the intended application of the patent was the bioethanol industry, which concerns itself primarily with ethanol production. Brewing applications must also consider the resulting organoleptic properties of the fermented wort. Controlled mixed-culture fermentations where common brewing strains of *S. cerevisiae* are paired with non-conventional species of yeast have attracted attention for the ability to reduce the ethanol content of model beers whilst enhancing the content of yeast-derived volatile compounds [17,18]. Non-conventional yeast species generally produce less ethanol than the highly adapted *S. cerevisiae* strains used in brewing, which tends to result in lower overall yields in mixed culture. Highly attenuative non-conventional strains may be used for the opposite purpose.

*Schizosaccharomyces pombe* is highly osmotolerant [14] and has previously been highlighted for its high fermentative power (i.e., ethanol production) both in brewing applications [19] and in winemaking [20], outperforming strains of *S. cerevisiae* species employed in both cases. This species of yeast is of specific oenological interest due to its ability to convert malolactic acid into lactic acid [21], and to reduce the content of the potentially carcinogenic compound ethyl carbamate [22]. Brewing-relevant advantages explored elsewhere include positive impacts on foam stability [19] and the production of low-carbohydrate beer [23]. This study aims to probe the potential of this yeast in lab-scale mixed fermentations of both standard and high-gravity model wort.

## 2. Materials and Methods

### 2.1. Strains

The strains utilised in this study were the commercially available *S. cerevisiae* strain LalBrew Nottingham™ (Lallemand Inc., Montreal, QC, Canada) and the *S. pombe* strain NCYC 535 (equivalent to CBS 4100), available from the National Collection of Yeast Cultures (Norwich, UK).

### 2.2. Yeast Isolation and Storage

Initially, lyophilised yeast strains were rehydrated in Universal Medium for Yeasts, composed of 3.0 g/L yeast extract (Oxoid, Basingstoke, UK), 3.0 g/L malt extract (Oxoid), 5.0 g/L bacteriological peptone (Oxoid), and 10.0 g/L glucose (Tereos, Moussy-le-Viex, France). The resulting yeast suspension was streaked on Universal Medium Agar, prepared as above but with 15.0 g/L agar (Oxoid), and incubated for 3 days at 28 °C. Large, well-separated colonies were chosen for inoculation into 5 mL yeast peptone dextrose (YPD) media (27 °C, 24 h, 180 rpm). The resulting yeast suspension was then mixed at a 1:1 ratio with sterile aqueous glycerol solution (30% *v/v*) and stored at −80 °C for long-term storage. For short-term storage, colonies were streaked on slants of YPD agar and incubated for 48–72 h until covered with a film of yeast. These were sealed with Parafilm and stored at 4 °C until required.

### 2.3. Wort Production (Standard-Gravity Wort)

Twenty L of wort was produced in a Grainfather G30 brewing system (Bevie Handcraft NZ Ltd., Auckland, NZ, USA). In brief, 5.8 kg of Finest Pale Ale Maris Otter Malt (Simpsons, Berwick-Upon-Tweed, UK) was milled in a M5 2-roller mill (Fraser Agricultural, Inverurie, UK), then mashed in at a 2.7:1 liquor-to-grist ratio, followed by a single-step infusion mash at 67 °C. A total 50 g of Fuggles hops (4.2% α-acid *w/w*, Charles Faram, Newland, UK) were added at the start of a 60 min boil, resulting in 20 L of hopped wort of a specific gravity (SG) of 1.051 or 12.6 g dry extract per 100 mL wort (°P) and pH 5.40. This was stored at −20 °C until required.

### 2.4. Wort Production (High-Gravity Wort)

Wort was produced under the same conditions as the standard-gravity wort but using the custom-built 50 L brewing kit at the International Centre for Brewing and Distilling (ICBD), resulting in a wort of 1.053 SG at 45 min; to this, 2.3 kg of a 1:2:3 mixture of glucose:maltodextrin:maltose was added, resulting in a wort of 1.093 SG (22.3 °P) and pH 5.40. This was stored at −20 °C until required.

### 2.5. Propagation of Yeast Cultures

A loopful of yeast was taken from a YPD agar slant and inoculated into 100 mL sterile YPD. This was stirred using a small (13 mm × 3 mm Ø) magnetic bar (1200 rpm) over the course of 72 h at 27 °C. The spent media was separated from the yeast pellet by centrifugation (1789 relative centrifugal force or RCF), and the pellet was resuspended in 100 mL fresh YPD and cultured for a further 24 h under the same conditions. The resulting culture was centrifuged a second time (1789 RCF) and resuspended in 20 mL sterile phosphate buffered saline (PBS) solution. The cell concentration and viability was measured by staining with Methylene Blue in a haemocytometer (Optik Labor, Görlitz, Germany) in line with European Brewery Convention (EBC) method 3.2.1.1 [24]. Cell viabilities above 99% were deemed fit for inoculation.

### 2.6. Inoculation of Wort

Wort was passively defrosted at room temperature (20–21 °C), coarsely filtered, and autoclaved at 115 °C and 10 pounds per square inch (psi) for 15 min. Once cooled, this wort was aerated using a sintered steel carbonation stone for 90 s at 20 °C. This was found to achieve a dissolved oxygen concentration of 6.92 ± 0.24 ppm (CarboQC ME, Anton Paar, Graz, Austria). The oxygenated wort was inoculated using the PBS-suspended yeast pellet to an initial total cell concentration of $1 \times 10^7$ viable cells/mL, according to the relative proportion of each strain in the mixed culture (20:1, 1:1, or 1:20 of *S. cerevisiae:S. pombe*) or monoculture. Inoculated wort was adjusted with PBS to maintain constant volume and wort strength (1.049 SG and pH 5.4). Fermentations of 50 mL inoculated wort were carried out in triplicate at 20 °C with stirring (13 mm × 3 mm Ø, 900 rpm).

### 2.7. Monitoring and Modelling of Fermentation Kinetics

The mass of each fermentation flask was taken at inoculation and at regular intervals for seven days, covering a minimum period of 168 h and comprising 14 readings per flask. A linear adjustment was applied to correct for evaporation of water from the airlocks over time, and the remaining mass loss was assumed to be due to $CO_2$ production. Modelling of the cumulative mass loss was performed by minimisation of the residual sum of squares (RSS) of the data to the model fit using the R software package (R-4.0.3, R Foundation for Statistical Computing, Vienna, Austria), adapting a script compiled by Reid and colleagues [25]. Modified 4- and 5-parameter logistic curves were compared using the corrected Akaike Information Criterion (AICc) to select for the best model [26].

*2.8. Determination of Free Amino Nitrogen (FAN) Content*

The EBC method 10.8.1 Free Amino Nitrogen in Wort by Spectrophotometry [27] was followed to find the FAN concentration of the samples. In essence, samples were diluted to contain 1–3 mg/L amino nitrogen and heated in the presence of ninhydrin reagent. After cooling, dilution, and quenching, the absorbance at 570 nm was taken. The relative absorbance of a stock glycine sample then allowed for the FAN concentration to be calculated.

*2.9. Alcohol by Volume (ABV), pH, Specific Gravity (SG), Extract, and Real Degree of Fermentation (RDF)*

ABV and SG were determined in a Beer ME Alcoholyzer (Anton Paar, Graz, Austria) and a DMA 4500 M Density Meter (Anton Paar). The pH was measured at 20 °C with an HI 83,141 pH meter (Hanna Instruments, Woonsocket, RI, USA) equipped with an HI1230 electrode (Hanna Instruments). Extract and RDF were calculated using the EBC polynomial relating SG to °P [28], but reported to one decimal place, and the EBC method 9.5 [29], respectively.

*2.10. Determination of Yeast Metabolite Concentration*

Yeast metabolites were determined in a 7820A gas chromatography (GC) system (Agilent, Santa Clara, CA, USA) coupled with a flame ionisation detector (FID). The system was equipped with a 7697A headspace injector (Agilent). A six-point calibration curve was constructed within the following concentration ranges for the measured volatiles: acetaldehyde (0.5–60 ppm); ethyl acetate (0.5–60 ppm); methanol (2.5–250 ppm); propan-1-ol (0.5–60 ppm); isobutanol (0.5–60 ppm); isoamyl acetate (0.3–35 ppm); butan-1-ol (1.8–340 ppm); isoamyl alcohol (1.8–340 ppm); ethyl hexanoate (0.01–1.8 ppm); ethyl octanoate (0.01–1.8 ppm); 2-phenylethyl acetate (0.015–2.5 ppm); 2-phenylethyl alcohol (1.1–225 ppm); ethyl lactate (2.4–470 ppm). Samples were adjusted to 9.5% ABV and heated to 85 °C for 15 min before injection through an Agilent DB-wax column (30 m × 25 mm internal diameter × 25 μm coating thickness) using hydrogen as a carrier gas. The injector and detector temperatures were, respectively, 120 °C and 270 °C. The sample was held initially at 30 °C for 6 min followed by ramping up to 60 °C (5 °C/minute) and holding for 2 min. This was ramped up to 210 °C (20 °C/minute) with no hold, and then to 250 °C (70 °C/minute) and held for one minute.

*2.11. Statistical Analysis*

Analytical data were subjected to Analysis of Variance (ANOVA) to check for significance, and means were compared where difference was found via Tukey's Honest Significant Difference (HSD) test using the SPSS package (IBM Corp. Released 2020. IBM SPSS Statistics for Windows, Version 27.0. Armonk, NY, USA: IBM Corp).

**3. Results**

*3.1. Fermentation of Standard-Gravity Wort by S. cerevisiae (SCN) and S. pombe (SZ1) Monocultures, and Three Mixed Cultures at 20:1, 1:1, and 1:20 Initial Inoculation Ratio*

Measuring the weight loss of the fermentation vessels is a non-invasive method for monitoring fermentation progress [30,31] and, thus, a useful method for the study of smaller fermentation volumes. Industrially, it is uncommon to autoclave wort before inoculation; however, autoclaved wort has been found to have similar physical and sensory characteristics to boiled wort [32] and was thus favoured for the increased control over microbial activity it offers. The outcomes of mixed-culture fermentations have been shown to depend strongly upon the ratio of *S. cerevisiae* to the non-conventional strain employed, where higher rates of the brewer's yeast can lead to higher ethanol content [17], and higher levels of volatiles [33] in some cases; mixed cultures are elsewhere shown to be uniquely capable of producing (for example) terpenoid esters [34]. Whilst these results are demonstrably strain-specific, the standard-gravity trial fermentations offer a useful insight

into each culture's performance at higher wort strength. All five fermentations were found to lose approximately the same mass (of $CO_2$) over the seven-day trial period (Figure 1); the first 80 h of the 168 h are shown for clarity. Also evident is a delay in the onset of fermentation in the fermentations at 20:1 and 0:1 (SCN:SZ1) inoculation ratios. The three cultures with higher relative viable cell counts of *S. cerevisiae* in the initial inoculum strongly resemble each other; higher proportions of *S. pombe* in the inoculum lead to longer lag times and a longer time until the exponential phase is concluded (Table 1). It is also the case that the maximum specific rate of fermentation is proportional to *S. pombe* inclusion rates. This faster maximum fermentation rate leads to the modelled exponential phase for *S. pombe* in monoculture ending 2.6 h after that recorded for the *S. cerevisiae* monoculture (34.6 ± 1.0 vs. 32.0 ± 0.8 h), despite a difference of 5.6 h for lag time (18.1 ± 0.3 vs. 12.5 ± 0.3 h). This trend is reflected in the kinetics of the mixed fermentations too.

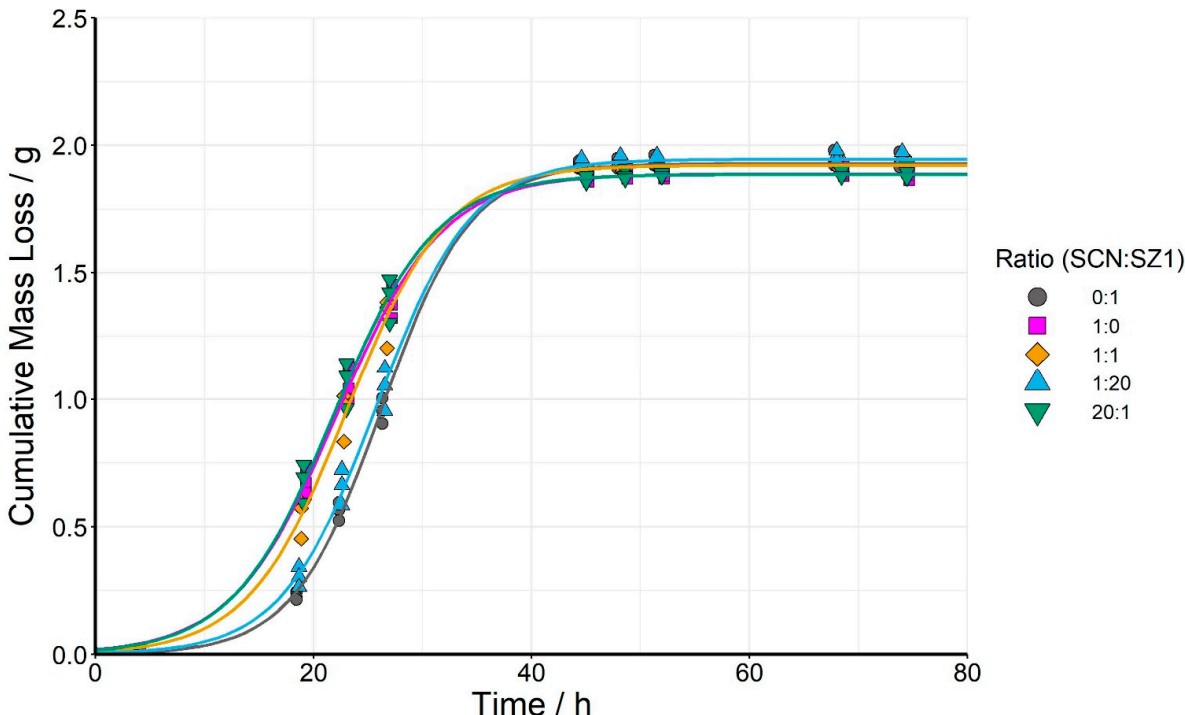

**Figure 1.** Fermentation progress over 80 h of *S. cerevisiae* (SCN) and *S. pombe* (SZ1) monocultures, and three mixed cultures at 20:1, 1:1, and 1:20 initial inoculation ratio in standard-gravity wort (1.049), modelled with the modified 4-parameter logistic curve. All readings from independent biological replicates (*n* = 3) are shown, with the model fit to the full set of data points.

**Table 1.** Fermentation kinetics of *S. cerevisiae* (SCN) *and S. pombe* (SZ1) monocultures, and three mixed cultures at 20:1, 1:1, and 1:20 initial inoculation ratio in standard-gravity wort (SG 1.049).

| Ratio (SCN:SZ1) | Lag Phase (h) | End of Exponential Phase (h) | Maximum Rate of Fermentation (g/h) |
|---|---|---|---|
| 1:0 | 12.6 ± 0.3 [a] | 31.7 ± 0.7 [ab] | 0.099 ± 0.003 [a] |
| 20:1 | 12.6 ± 0.7 [a] | 31.2 ± 1.0 [a] | 0.102 ± 0.003 [ab] |
| 1:1 | 14.1 ± 0.7 [a] | 32.1 ± 1.5 [ab] | 0.107 ± 0.004 [bc] |
| 1:20 | 17.1 ± 0.7 [b] | 34.4 ± 1.0 [b] | 0.113 ± 0.001 [cd] |
| 0:1 | 18.1 ± 0.3 [b] | 34.4 ± 0.8 [b] | 0.119 ± 0.002 [d] |

Mean values of independently modelled data series (*n* = 3) shown ± SD. Mean values not sharing a letter within a column were found to be statistically different according to Tukey's HSD test (*p* < 0.05).

The *S. pombe* monoculture was able to produce a fermented wort with characteristics similar to that of the *S. cerevisiae* monoculture; the model beer was found to have similar

final pH values and ethanol content in both cases (Table 2). Accordingly, the three mixed fermentations were also found to have a similar ABV. However, at 1:1 inoculation ratio, the pH was found to be significantly lower than that produced by either monoculture, and close to the point (i.e., a pH of <3.9) where it might be considered "sour beer" [35]. The final gravity (FG) was found to be significantly lower at both 1:1 and 1:20 inoculation ratio, although it is unknown whether the difference in sweetness and mouthfeel between any two of the model beers would be perceived. *S. pombe* in monoculture was also found to use less FAN to achieve a similar degree of fermentation, which would be beneficial for the fermentation of nitrogen-deficient worts. This apparent nitrogen efficiency was not observed in mixed culture fermentation.

**Table 2.** Fermented wort characteristics of *S. cerevisiae* (SCN) and *S. pombe* (SZ1) monocultures, and three mixed cultures at 20:1, 1:1, and 1:20 initial inoculation ratio in standard-gravity wort (SG 1.049).

| Ratio (SCN:SZ1) | Residual FAN (mg/L) | FAN Consumption (%) | pH | FG | RDF (%) | ABV (*v/v* %) |
|---|---|---|---|---|---|---|
| 1:0 | 49 ± 1 [a] | 76.3 ± 0.6 [b] | 4.16 ± 0.03 [b] | 1.00632 ± 0.00003 [c] | 84.8 ± 0.2 [a] | 5.56 ± 0.08 [a] |
| 20:1 | 44 ± 1 [a] | 78.7 ± 0.4 [b] | 4.08 ± 0.02 [ab] | 1.00516 ± 0.00008 [b] | 87.3 ± 0.2 [b] | 5.59 ± 0.02 [a] |
| 1:1 | 46 ± 4 [a] | 77.7 ± 2.0 [b] | 3.93 ± 0.11 [a] | 1.00443 ± 0.00007 [a] | 89.1 ± 0.2 [d] | 5.74 ± 0.12 [a] |
| 1:20 | 47 ± 5 [a] | 77.5 ± 2.3 [b] | 4.21 ± 0.04 [b] | 1.00442 ± 0.00015 [a] | 88.9 ± 0.1 [d] | 5.62 ± 0.13 [a] |
| 0:1 | 62 ± 2 [b] | 70.2 ± 0.9 [a] | 4.12 ± 0.03 [b] | 1.00496 ± 0.00010 [b] | 87.8 ± 0.2 [c] | 5.65 ± 0.01 [a] |

Mean values of independent biological replicates (*n* = 3) shown ± SD. Mean values not sharing a letter within a column were found to be statistically different according to Tukey's HSD test (*p* < 0.05). FAN = Free Amino Nitrogen; FG = Final Gravity; RDF = Real Degree of Fermentation; ABV = Alcohol By Volume.

A set of volatile organic compounds was quantified by HS-GC-FID analysis. In general, concentrations of these compounds were found to be broadly similar across all five cultures, with the exception of isobutanol and isoamyl alcohol, which were found in their highest concentrations at 1:1 inoculation ratio (Table 3). Isoamyl alcohol, which imparts an alcoholic and banana-like aroma, was detected in quantities at or above the aroma threshold (50–65 ppm) at 1:1 ratio, as was its acetate ester (1.2–2 ppm) [36] at all ratios studied. The production of isoamyl acetate by brewing yeast is thought to depend on the uptake of leucine during fermentation [37]. Leucine metabolism in *S. cerevisiae* is controlled by the *BAP2* gene, which mediates branched-chain amino acid permeases [38], and the *BAT1* gene, which mediates the activity of branched-chain amino acid transferases [39]. Although most of the compounds were not detected in quantities above established aroma thresholds, they may still be evident in the overall aroma of the beer through synergistic effects [40,41].

**Table 3.** Volatile organic compounds in fermented wort inoculated with *S. cerevisiae* (SCN) and *S. pombe* (SZ1) monocultures, and three mixed cultures at 20:1, 1:1, and 1:20 initial inoculation ratio in standard-gravity wort (1.049).

| Ratio (SCN:SZ1) | 1:0 | 20:1 | 1:1 | 1:20 | 0:1 | Aroma Threshold (mg/L) | |
|---|---|---|---|---|---|---|---|
| Compound | | | Concentration (mg/L) | | | | Reference |
| Acetaldehyde | 3.15 ± 1.10 [a] | 3.42 ± 0.22 [a] | 22.0 ± 20.6 [a] | 20.6 ± 25.7 [a] | 21.0 ± 4.0 [a] | 25 | [42] |
| Methanol | 7.50 ± 0.98 [a] | 6.95 ± 0.64 [a] | 28.1 ± 22.6 [a] | 8.01 ± 4.27 [a] | 4.14 ± 0.86 [a] | 10,000 | [43] |
| Propan-1-ol | 17.9 ± 4.2 [b] | 9.51 ± 1.86 [ab] | 6.00 ± 8.01 [ab] | 4.88 ± 4.23 [a] | ND [a] | 600 | [36] |
| Butan-1-ol | 0.151 ± 0.023 [a] | 0.150 ± 0.015 [a] | 0.203 ± 0.066 [a] | 0.136 ± 0.028 [a] | 0.0807 ± 0.0699 [a] | 450 | [43] |
| Isobutanol | 19.7 ± 2.0 [b] | 17.8 ± 1.0 [b] | 31.4 ± 5.2 [c] | 20.2 ± 4.4 [b] | 1.75 ± 0.27 [a] | 100 | [36] |
| Isoamyl alcohol | 36.2 ± 4.3 [b] | 31.4 ± 1.9 [b] | 64.2 ± 11.6 [c] | 40.5 ± 5.4 [b] | 6.13 ± 1.37 [a] | 50–65 | [36] |
| 2-phenylethyl alcohol | ND [a] | ND [a] | 26.0 ± 24.2 [a] | ND [a] | ND [a] | 40 | [36] |
| Isoamyl acetate | 2.95 ± 0.33 [a] | 3.39 ± 0.19 [ab] | 3.91 ± 0.30 [b] | 3.21 ± 0.07 [a] | 3.83 ± 0.15 [b] | 1.2–2.0 | [36] |
| Ethyl acetate | 3.54 ± 0.99 [a] | 2.35 ± 0.29 [a] | 4.36 ± 1.02 [a] | 4.46 ± 2.55 [a] | 4.80 ± 1.50 [a] | 25–30 | [36] |
| Phenylethyl acetate | ND | ND | ND | ND | ND | 0.2–3.8 | [36] |
| Ethyl hexanoate | 0.102 ± 0.053 [a] | 0.143 ± 0.071 [a] | 0.0835 ± 0.0258 [a] | 0.0572 ± 0.0135 [a] | 0.147 ± 0.076 [a] | 0.2–0.23 | [36] |
| Ethyl octanoate | 0.0364 ± 0.0235 [a] | 0.0673 ± 0.0203 [a] | 0.122 ± 0.105 [a] | 0.0357 ± 0.0219 [a] | 0.0468 ± 0.0031 [a] | 0.9–1.0 | [36] |
| Ethyl lactate | 0.954 ± 0.831 [a] | 1.16 ± 1.07 [a] | 1.14 ± 1.97 [a] | 0.104 ± 0.179 [a] | 1.11 ± 1.14 [a] | 154 | [43] |

Mean values of independent biological replicates (*n* = 3) shown ± SD. Mean values not sharing a letter within a row were found to be statistically different according to Tukey's HSD test (*p* < 0.05). ND = Not Detected.

### 3.2. Fermentation of High-Gravity Wort by S. cerevisiae (SCN) and S. pombe (SZ1) Monocultures and by Mixed Culture at 1:1 Initial Inoculation Ratio

The *S. cerevisiae* and *S. pombe* strains were trialled in monoculture and in mixed fermentation at a 1:1 initial inoculation ratio in a higher-gravity wort. In standard-gravity wort, potentially synergistic overproduction of volatiles was found only at this inoculation ratio, and the intermediate mixed cultures (20:1 and 1:20) offered no clear kinetic advantage over the 1:1 mixed culture or monocultures. For these reasons, these were not studied in stronger worts. The wort was similar to the wort produced for the trials at standard gravity but with 30% of the sugar content derived from a mixture of glucose, fructose, and maltodextrin to simulate the monosaccharide, disaccharide, and trisaccharide/dextrin fractions found in high maltose syrup [1]. The three cultures display clearly different fermentation kinetics (Figure 2): the *S. cerevisiae* monoculture was the most active of the three over the first 30 h of the fermentation period but plateaued at around 3.1 g of mass lost from the 50 mL miniature fermentation vessels.

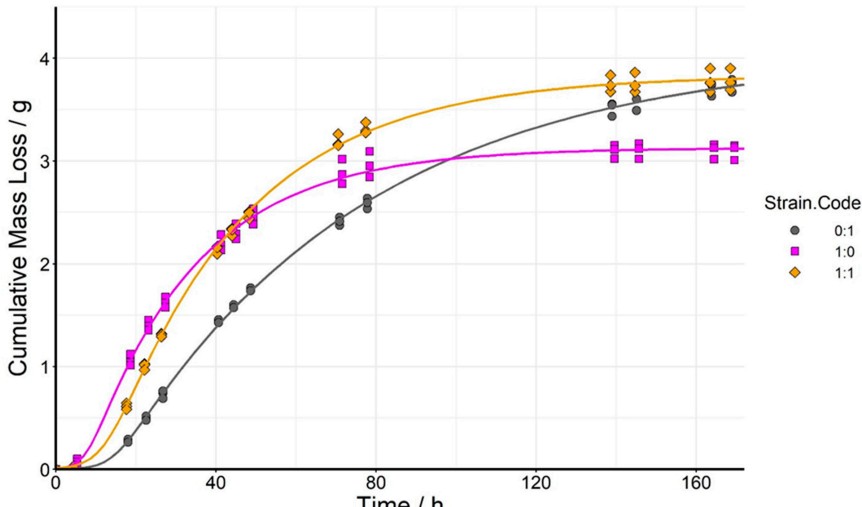

**Figure 2.** Fermentation progress over 168 h of *S. cerevisiae* (SCN) and *S. pombe* (SZ1) monocultures, and mixed culture at 1:1 initial inoculation ratio in high-gravity wort (1.088 SG). All readings from independent biological replicates (*n* = 3) are shown, with the model fit to the full set of data points.

The *S. pombe* monoculture produced the least $CO_2$ until almost the 5th day of the fermentation period and did not reach its asymptote during this time. The mixed culture produced the most $CO_2$ over the time period and had intermediate kinetics (Table 4). *S. cerevisiae* had the shortest modelled lag time, the shortest time elapsed until the end of the exponential phase, and the fastest rate of fermentation. The opposite is true of the *S. pombe* monoculture, and the mixed culture was intermediate between these two. In percentage terms, the differences are quite large and would certainly impact the number of batches that could be produced in a commercial brewing facility.

**Table 4.** Fermentation kinetics of *S. cerevisiae* (SCN) and *S. pombe* (SZ1) monocultures, and mixed culture at 1:1 initial inoculation ratio in high-gravity wort (1.088).

| Ratio (SCN:SZ1) | Lag Time (h) | End of Exponential Phase (h) | Maximum Rate of Fermentation (g/h) |
|---|---|---|---|
| 1:0 | 6.3 ± 0.3 [a] | 41.7 ± 0.8 [a] | 0.088 ± 0.001 [c] |
| 1:1 | 10.1 ± 0.4 [b] | 57.0 ± 1.7 [b] | 0.082 ± 0.001 [b] |
| 0:1 | 13.1 ± 0.4 [c] | 87.8 ± 1.9 [c] | 0.054 ± 0.000 [a] |

Mean values of independently modelled data series (*n* = 3) shown ± SD. Mean values not sharing a letter within a column were found to be statistically different according to Tukey's HSD test (*p* < 0.05).

The *S. pombe* monoculture was found to have a similar pH, FG, RDF, and ABV to the mixed culture after seven days, whilst the *S. cerevisiae* had a higher pH and FG, and a lower ABV (Table 5). The pH of the wort produced by the *S. cerevisiae* was similar to that reported elsewhere in HG fermentations [44], but the mixed culture and *S. pombe* monoculture acidified the wort considerably, to pH 3.55 and 3.43, respectively. Although low pH (respective to *S. cerevisiae*) has previously been reported for this strain in single culture [19] for wort fermentations, elsewhere relatively higher pH was found for cider fermentations [45]. In a mixed *S. pombe–S. cerevisiae* wine fermentation, concentrations of organic acids were found to be lower (in the case of malic acid), slightly higher than (lactic acid), or equal to (tartaric acid) the model wine produced by *S. cerevisiae* alone [46]. It is unknown whether this lower pH is a simple consequence of increased fermentative activity or a consequence of the production of organic acids by this species. The higher ABV and lower FG indicate that this strain is either more ethanol tolerant than the *S. cerevisiae* strain in the trial, or that it can ferment wort components (such as trisaccharides or dextrins) that *S. cerevisiae* cannot. This *S. pombe* strain is reported to be able to utilise soluble starch under aerobic conditions by the National Collection of Yeast Cultures and trials are underway to independently assess the strain's diastatic activity. In any case, a separate experiment screening yeast strains for the phenolic off-flavour (POF) phenotype found this strain to be POF-, in contrast to most diastatic strains of *S. cerevisiae* [47]. Both the *S. pombe* strain and the mixed culture may then find use in the production of high-ABV beer, where the POF phenotype is not desired, as an alternative strategy to rare mating [48] or hybridisation [49] strategies attempted elsewhere.

**Table 5.** Fermented wort characteristics of *S. cerevisiae* (SCN) and *S. pombe* (SZ1) monocultures, and mixed culture at 1:1 initial inoculation ratio in high-gravity wort (1.088).

| Ratio (SCN:SZ1) | Residual FAN (mg/L) | FAN Consumption (%) | pH | FG | RDF (%) | ABV (*v/v* %) |
|---|---|---|---|---|---|---|
| 1:0 | 46 ± 1 [a] | 77.7 ± 0.4 [a] | 3.91 ± 0.07 [b] | 1.02308 ± 0.00057 [b] | 69.8 ± 0.5 [a] | 8.46 ± 0.03 [a] |
| 1:1 | 41 ± 2 [a] | 80.0 ± 1.1 [a] | 3.55 ± 0.02 [a] | 1.00975 ± 0.00051 [a] | 87.0 ± 0.6 [b] | 10.32 ± 0.02 [b] |
| 0:1 | 47 ± 1 [a] | 77.1 ± 0.3 [a] | 3.43 ± 0.01 [a] | 1.00858 ± 0.00220 [a] | 88.5 ± 2.7 [b] | 10.45 ± 0.09 [b] |

Mean values of independent biological replicates ($n = 3$) shown ± SD. Mean values not sharing a letter within a column were found to be statistically different according to Tukey's HSD test ($p < 0.05$). FAN = Free Amino Nitrogen; FG = Final Gravity; RDF = Real Degree of Fermentation; ABV = Alcohol By Volume.

The aroma-active volatile compounds produced during fermentation of HG wort were, in general, found to be in higher concentration than those produced in standard wort (Table 6). In particular, isoamyl acetate, isoamyl alcohol, ethyl hexanoate, and phenylethyl acetate were all found to be in higher concentration than their reported aroma threshold for at least one of the model beers. Notably, the mixed culture produced significantly higher quantities of isoamyl acetate, isoamyl alcohol, and phenylethyl acetate than either of the monocultures, as well as methanol and n-butanol. The impact of the fermentation of HG worts on ester/higher alcohol production depends on the nature of the additional extract. The use of all-malt HG wort has been found to increase the concentration of isoamyl acetate, amyl alcohol, propanol, and isobutanol compared with the fermentation of both a standard-gravity wort and an HG wort supplemented with very-high-maltose (VHM) syrup [50]; the use of glucose as an adjunct in HG wort has elsewhere been shown to increase the concentration of isoamyl alcohol and its acetate ester [44]. Thus, with careful consideration of both the strength and composition of the wort alongside the fermentation culture, brewers may produce beers with desirable organoleptic properties and fermentation performance.

**Table 6.** Volatile organic compounds detected by GC-FID for SCN and SZ1 in monoculture and in mixed (1:1) culture high-gravity wort.

| Ratio (SCN:SZ1) | 1:0 | 1:1 | 0:1 | Aroma Threshold (mg/L) | |
|---|---|---|---|---|---|
| Compound | Concentration (mg/L) | | | | Reference |
| Acetaldehyde | 9.98 ± 8.58 [a] | 6.81 ± 1.43 [a] | 7.68 ± 1.07 [a] | 25 | [42] |
| Methanol | 45.9 ± 7.4 [a] | 66.0 ± 8.2 [b] | 42.6 ± 3.2 [a] | 10,000 | [43] |
| Propan-1-ol | ND [a] | ND [a] | 4.77 ± 5.80 [a] | 600 | [36] |
| Butan-1-ol | 11.1 ± 0.0 [a] | 12.8 ± 0.5 [b] | 11.8 ± 0.2 [a] | 450 | [43] |
| Isobutanol | 62.3 ± 5.1 [b] | 59.4 ± 8.9 [b] | 18.1 ± 1.6 [a] | 100 | [36] |
| Isoamyl alcohol | 145 ± 11 [b] | 280 ± 45 [c] | 30.7 ± 2.6 [a] | 50–65 | [36] |
| 2-phenylethyl alcohol | ND | ND | ND | 40 | [36] |
| Isoamyl acetate | 1.58 ± 0.02 [a] | 2.06 ± 0.11 [b] | 1.54 ± 0.02 [a] | 1.2–2.0 | [36] |
| Ethyl acetate | 6.80 ± 1.34 [a] | 9.07 ± 1.37 [a] | 6.18 ± 1.09 [a] | 25–30 | [36] |
| Phenylethyl acetate | 0.140 ± 0.016 [a] | 0.369 ± 0.062 [b] | 0.0394 ± 0.0682 [a] | 0.2–3.8 | [36] |
| Ethyl hexanoate | 0.194 ± 0.030 [a] | 0.210 ± 0.066 [a] | 0.150 ± 0.006 [a] | 0.2–0.23 | [36] |
| Ethyl octanoate | 0.159 ± 0.001 [a] | 0.174 ± 0.003 [b] | 0.171 ± 0.005 [b] | 0.9–1.0 | [36] |
| Ethyl lactate | 11.7 ± 10.4 [a] | 7.73 ± 13.40 [a] | 5.52 ± 9.57 [a] | 154 | [43] |

Mean values of independent biological replicates ($n$ = 3) shown ± SD. Mean values not sharing a letter within a row were found to be statistically different according to Tukey's HSD test ($p < 0.05$). ND = Not Detected.

## 4. Conclusions

Mixed cultures may be a further tool to modulate and adjust the pre-dilution concentration of aroma-active volatiles to fit the brand profile. Efficiency savings from HG brewing are found once the fermented wort is diluted, and consideration of this aspect is informative. Normalising the volatile organic compound concentrations to a model beer of 5.16% ABV (i.e., diluting the mixed culture 50% with water) would reduce the concentration of ethyl hexanoate to around half of its threshold value, and leave isoamyl acetate and phenylethyl acetate just below theirs; isoamyl alcohol would remain well above its threshold value. Diluting to the same ABV for the *S. cerevisiae* monoculture would result in 18% less additional product and would have a similar impact on the volatile profile. The feedstock (wort composition) and culture must then be carefully selected to produce a beer with the desired properties to take advantage of the efficiency savings inherent in HG brewing with mixed cultures.

The use of mixed-culture fermentation offers another path to modify the fermentation kinetics, aroma profile, and ethanol yield of high-gravity fermentations. Mixed cultures may represent compromise positions between yeast strains with faster or slower kinetics and may also demonstrate synergistic increases in the production of aroma-active volatile compounds. The exploitation of non-*Saccharomyces* enzymatic machinery may lead to the discovery of commercially useful phenotypes such as POF- diastatic activity, which could offer brewers new tools to achieve their objectives.

**Author Contributions:** Conceptualization, D.J.; Investigation, B.P.; Methodology, B.P. and S.J.R.; Supervision, A.E.H. and D.J.; Writing—original draft, B.P.; Writing—review & editing, A.E.H. and D.J. All authors have read and agreed to the published version of the manuscript.

**Funding:** This research received no external funding.

**Conflicts of Interest:** The authors declare no conflict of interest.

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
