# Peer review of "Schizosaccharomyces pombe in the Brewing Process: Mixed-Culture Fermentation for More Complete Attenuation of High-Gravity Wort"

_fermentation, doi:10.3390/fermentation8110643_

Round 1

Reviewer 1 Report

The manuscript interesting, well written, well organized and results are statistically analyzed. Before it can be considered for publication, there are some points that need attention:

·       Item 2.1: were the strains purchased in the lyophilized form? If so, please detail how it was activated to be streaked in agar plates
·       Please provide references for items 2.3 and 2.4

·       Industrially, wort for beer production is not autoclaved. How authors know that the sterilization process did not degrade sugars and nutrients?

·       Item 2.6: what was the volume of fermentation assays? And please provide a reference for this methodology. In addition, the co-culture and monoculture need to be detailed, such as inoculation co-culture rate etc.

·       Line 133: “(Reid et al., 2021)” is not in accordance with journal guidelines format for references. Please correct it.

·       Figure 1 and 2 – please add deviation bars to the experimental points

·       Table 2 – please add abbreviations meaning in the footnote

·       What were cells viabilities in the beginning of beer production? Does the magnetic bar used in the propagation step damage cells?

·       Please carefully check if all microorganisms’ names are in italic in the text

·       Please explain why authors choose 1:1 coculture proportion for high gravity experiments.

·       Line 261: please correct “(Error! Reference source not found.).

Reviewer 2 Report

The paper describes a straightforward experiment of co-inoculation with potential to be applied in the beer industry. The experimental design is standard and effective, with results presented in a structured manner. Although the new technological intervention is not sufficiently convincing to provide final products with desirable sensory quality, as a preliminary result, the paper is worth publishing.

The introduction chapter is clear, containing the arguments for conducting the present research.

The materials and methods, including monitoring and modelling of kinetics, as well as determination of yeast metabolites, are thoroughly exposed. Experiments are correctly run in triplicate and appropriate statistical analyses applied.

The results are sequentially presented, for the fermentation of standard gravity wort and then for the high gravity wort. The more complex design for standard gravity wort was reduced to a simpler one for high gravity wort. In the paper, it is not clearly explained, however, why in the case of the high gravity wort the ratios 1:20 and 20:1 were not performed anymore, especially when the title implies that is a research targeting the optimisation of high gravity wort fermentation. At least a short explanation about this would be desirable.

As the title and conclusions are only about high gravity wort fermentation, clearer justifications should be given to explain why the results for standard gravity wort fermentation were included in the paper. If the phrase in lines 75-76 “This study aims to probe the potential of this yeast in lab-scale mixed fermentations of both standard and high-gravity model wort.” is to be maintained, than, the title should be modified to also include standard gravity wort.

Some other minor revisions:

-       -   In lines 239 and 261 two reference numbers are missing in the text, being incorrectly cited as “(Error! Reference 239 source not found.)”

-     -     The abbreviations of all determined parameters should be present in the paper. This is the case of “FG”, which is clear for the specialists, but not defined anywhere as being the final gravity. Could be enough if in line 198 “The final gravity was found” would become “The final gravity (FG) was found”. It should also be checked for other similar cases.

-      -    In the Tables, the significant different concentrations of volatiles were highlighted in grey. To be consistent, in table 6, for the ratio 1:1 the higher concentration of methanol should be highlighted, while the concentration of ethyl hexanoate, which is not statistically different than in the case of monocultures, should not be highlighted.

-     -     In Figure 1 the average curve of for the inoculation ratio 1:20 is not visible enough and needs to be drawn with a darker colour.  

Reviewer 3 Report

This work is aimed to study the suitability of two yeast starter, belonging to Schizosaccharomyces pombe and Saccharomyces cerevisiae species, for the production of high gravity beer. The overall presentation is clear and concise. The experimental approach is adequate for the purpose, materials and methods used to perform different assays are properly described and statistical treatment of resulting data is appropriate. Moreover, this study proved that industrial application of this mixed starter culture determined beers possessing with specific aromatic profiles. These findings might be interesting in  the application of mixed culture inoculation in high gravity brewing practices allowing, also the increase of brewery efficiency.

I have only a few minor points to improve the manuscript:
- Line 94: change "20" with "Twenty";
- Lines 239-240: check the reference;
- Line 260: check the reference;
- Table 6: please include two additional columns in the table: (i) the first where the perception threshold values found in the literature for each volatile organic compound are given; (ii) odour activity value (OAV) for each volatile organic compound.

Round 2

Reviewer 1 Report

The authors have attended all raised queries accordingly. Now I recommend the manuscript for publication.